# Arytenoid Fixation in Laryngeal Cancer: Radiological Pictures and Clinical Correlations with Respect to Conservative Treatments

**DOI:** 10.3390/cancers11030360

**Published:** 2019-03-13

**Authors:** Giovanni Succo, Stefano Cirillo, Ilaria Bertotto, Elena Maldi, Davide Balmativola, Massimo Petracchini, Dario Gned, Alessandro Fornari, Gian Marco Motatto, Andrea E. Sprio, Andrea Manca, Erika Crosetti

**Affiliations:** 1Head and Neck Oncology Service, Candiolo Cancer Institute—FPO IRCCS, 10060 Candiolo (TO), Italy; giovanni.succo@ircc.it (G.S.); andrea.manca@ircc.it (A.M.); 2Department of Oncology, University of Turin, 10043 Orbassano (TO), Italy; 3Radiology Unit, Mauriziano Umberto I Hospital, 10128 Turin, Italy; scirillo@mauriziano.it (S.C.); massimo.petracchini@gmail.com (M.P.); 4Radiology Unit, Surgery Department, Candiolo Cancer Institute, FPO—IRCCS, 10060 Candiolo (TO), Italy; ilaria.bertotto@ircc.it; 5Pathology Unit, Medicine Department, Candiolo Cancer Institute, FPO—IRCCS, 10060 Candiolo (TO), Italy; elena.maldi@ircc.it (E.M.); davide.balmativola@ircc.it (D.B.); 6Department of Diagnostic Imaging, San Luigi Gonzaga Hospital, University of Turin, 10043 Orbassano (TO), Italy; dario_gned@yahoo.it; 7Department of Oncology, Division of Pathology, San Luigi Gonzaga Hospital, University of Turin, 10043 Orbassano (TO), Italy; alefornari77@gmail.com; 8Otorhinolaryngology Unit, University of Turin, 10124 Turin, Italy; gian.motatto@edu.unito.it; 9Department of Clinical and Biological Sciences, University of Turin, 10043 Orbassano (TO), Italy; andrea.sprio@unito.it

**Keywords:** open partial horizontal laryngectomy, supracricoid partial laryngectomy, supratracheal partial laryngectomy, T3 laryngeal cancer, laryngeal cancer, fixed arytenoid

## Abstract

*Background:* The aim of this retrospective study was to identify different radiological features in intermediate–advanced laryngeal cancer (LC) associated with arytenoid fixation, in order to differentiate cases still safely amenable to conservative treatment by partial laryngectomy or chemoradiotherapy. *Methods*: 29 consecutive patients who underwent open partial horizontal laryngectomies (OPHLs), induction chemotherapy followed by radiotherapy in the case of >50% response (IC + RT) or total laryngectomy were classified as: pattern I (supraglottic LC fixing the arytenoid due to weight effect), pattern II (glottic LC involving the posterior paraglottic space and spreading toward the crico-arytenoid joint and infraglottic extension <10 mm), pattern III (glottic—infraglottic LC involving the crico-arytenoid joint and infraglottic extension >10 mm) and pattern IV (transglottic and infraglottic LC with massive crico-arytenoid unit involvement, reaching the hypopharyngeal submucosa). All glottic cancers treated with surgery were studied by a cross sectional approach. *Results:* A substantial agreement between the work-up and the pathology results has been obtained in each of the subcategories. Three-year disease-free survivals, local control and freedom from laryngectomy were significantly better in pattern II compared to pattern III–IV. *Conclusions*: LC showing fixed arytenoid due to weight effect or posterior paraglottic space involvement with infraglottic extension <10 mm assessed at the true vocal cord midline are still safely manageable by OPHL or IC + RT.

## 1. Introduction

The selection of laryngeal cancer (LC), still manageable by OPHL, requires a very precise clinical-radiological work-up and an accurate evaluation of patient-related parameters. Among the tumor-related parameters, arytenoid fixation is one of the most delicate, as a possible negative prognostic factor for OPHLs.

Traditionally, in fact, arytenoid fixation was adopted as an exclusion criterion for partial laryngectomy [1,2,3], since it has been considered, as a minimum, the consequence of tumor extension to the thyro-arytenoid muscle (TAM), to the inferior paraglottic space (IPGS) and the crico-arytenoid joint (CAJ), if not to the lateral/posterior crico-arytenoid muscle and inferior laryngeal nerve [3,4]. Peritumoral inflammation, a frequent finding in the MRI study of LC involving the PGS, can also be considered as a co-factor in determining arytenoid fixation, due to crico-arytenoid arthritis [5]. Many authors have studied this phenomenon from the anatomic and pathological viewpoint. Derived from literature data, Figure 1 shows the rate of involvement of the structures composing the crico-arytenoid unit (CAU), which can cause arytenoid fixation, singly or in association [1,3,4,5,6,7,8,9,10,11,12].

In a study on 77 cases, Katilmis et al. focused on the different causes of arytenoid fixation: involvement of the intrinsic laryngeal muscle with insertion on the arytenoid, CAJ involvement and recurrent nerve infiltration. Another cause of reduced/absent arytenoid mobility is the weight effect, present in 60% of supraglottic tumors [4].

In earlier studies conducted on large series of patients with intermediate to advanced LC, our group have demonstrated that simple selection criteria can be applied to identify cases manageable by OPHL with a high degree of certainty [13].

This conservative surgical approach can be also be applied very carefully in selected cases of tumors reaching the inferior paraglottic space associated with arytenoid fixation. In fact, the subset of patients belonging to this subcategory comprises tumors characterized by biological behavior as well as oncologic results very similar to those of T4a cancers.

In this subcategory of tumors, characterized by insidious patterns of spread, it is still necessary to better clarify if and in which cases it is quite safe to use an OPHL, focusing on indications based on a study of different patterns of fixed arytenoid.

This retrospective mono-institutional study, conducted on a consecutive series of 29 patients affected by T3-T4a untreated LC, had as its unique objective the attempt to identify homogeneous subcategories in the large group of lesions determining arytenoid fixation, in order to understand which cases could be considered for conservative laryngeal treatment with sufficient certainty.

## 2. Patients and Methods

### 2.1. Patients

Twenty-nine consecutive untreated patients affected by laryngeal squamous cell carcinoma (LC) in the intermediate/advanced stage showing both vocal cord and arytenoid fixation and studied radiologically at our institutions, underwent uni/multimodal treatment between January 2012 and December 2015 at San Luigi Gonzaga Hospital and Candiolo Cancer Institute of Turin. In this retrospective study, all of the patients were in good general condition and wrote informed consent. The work-up was based on superficial and deep tumor extent, assessed by endoscopic and imaging evaluation performed less than 3 weeks before surgery. Computed tomography (CT) or magnetic resonance (MR) imaging was used to assess infiltration of the cartilaginous framework, involvement of the pre-epiglottic (PES) and/or paraglottic spaces (PGS), as well as extralaryngeal extension. Pre- and intraoperative videolaryngoscopic examination, by flexible endoscope in the office, and 0° and 70° rigid endoscopes in the operating theatre, were employed to evaluate both vocal cord/arytenoid mobility and superficial tumor extent. Patient characteristics, distribution according to the involved laryngeal sites, as well as their cT and cN categories are reported in Table 1.

At the time of diagnosis all of the patients showed an overall eligibility for the therapeutic options foreseen in AIOM guidelines for LC treatment in this stage [14]. Criteria for OPHL were histological diagnosis of intermediate/advanced (cT3 and cT4a who refused total laryngectomy) categories of glottic and supraglottic LC, Karnofsky index [15] higher than 80, cN0–N1. The absolute contraindications to OPHL type II and III, based on T and N stage were represented by: I) lesions extending to base of tongue or pyriform sinus; II) lesions with major invasion of the pre-epiglottic space involving the hyoid bone, lesions involving the inter-arytenoid space, the posterior commissure and both arytenoid cartilages; III) large extralaryngeal spread of cancer involving the thyroid gland, strap muscles, cervical skin, internal jugular vein or common carotid artery; IV) lesions reaching the first tracheal ring.

The suspected presence of clinically positive nodes >cN1 was not considered an absolute contraindication. However, it should not represent a good indication for OPHL due to the probable need for post-operative RT [13]. Criteria for induction chemotherapy followed by radiotherapy (IC + RT) were histological diagnosis of intermediate/advanced (cT3 and cT4a who refused total laryngectomy) categories of glottic and supraglottic LC, Karnofsky index higher than 80 and cN0–N3. Criteria for total laryngectomy (TL) were histological diagnosis of intermediate/advanced (cT3 and cT4a) categories of glottic and supraglottic LC, Karnofsky index higher than 70 and cN0–N3. 

### 2.2. Radiologic Assessment

Nineteen patients underwent just MRI, eight only a CT scan and two patients had both scans. All scans were evaluated by radiologists experienced in head and neck cross-sectional studies and were subsequently reviewed by two radiologists to identify a homogeneous pattern of arytenoid fixation. MRI and CT scans were executed to evaluate: pre-epiglottic space (PES), superior paraglottic space (SPGS), inferior paraglottic space (IPGS), crico-arytenoid joint (CAJ), thyroid cartilage (TC), cricoid ring, cricoid plate, infraglottic extension (measured from the free edge of the true vocal cord (TVC), at its midline), thyro-arytenoid muscle (TAM), arytenoid (ARY), crico-thyroid membrane (CAM), lateral crico-thyroid space (LCTS), extra-laryngeal extension (ELE). The MRI study has been conducted using Ingenia Healthcare equipment (Philips S.p.A., Milan, Italy) with 1.5 T magnet and using surface coils (24 channels). The study protocol included: T2-weighted images (FOV 16 cm, matrix 268 × 268, TR 3000 ms, TE 90 ms, slice thickness 3 mm) acquired on axial and coronal planes and T1-weighted images acquired on the axial plane (FOV 16 cm, matrix 512 × 256, TR 400 ms, TE 14 ms, slice thickness 3 mm). Diffusion weighted imaging (DWI) sequences have been acquired on the axial plane (FOV 25 × 25 cm, matrix 250 × 250, TR 2600 ms, TE 87 ms, slice thickness 3 mm) using spin-echo and echo-planar sequences, with B values of, respectively, 0, 500, 800, through which the corresponding apparent diffusion coefficient (ADC) map has been generated. At the end of the exam, after the injection of a paramagnetic contrast agent (Gadobutrol), T1-3D THRIVE axial (FOV 27 × 27 cm, matrix 448 × 364, TR 8.25, TE 3.16, slice thickness 0.6 mm) and T1-weighted coronal scans were acquired. All the scans were oriented according to the glottic plane.

CT images were acquired using a SOMATOM Definition Flash machine (Siemens Healthcare s.r.l., Milan, Italy). The protocol used was: 120 kV with automatic mAs modulation (using an iterative algorithm for reconstruction), slice thickness of 2 mm and reconstruction interval of 2 mm (with the possibility to reconstruct down to 0.6 mm) during iodine contrast injection.

### 2.3. Pattern of Arytenoid Fixation

Before treatment, based on clinical-radiological work-up, four different patterns of arytenoid fixation were identified. Comparison between the work-up and the pathology results was obtained by a cross-sectional study. All patients were divided into 4 patterns of arytenoid fixation. In all of the patients, the tumor involved the laryngeal compartments located posteriorly to a vertical virtual plane tangential to the arytenoid vocal process and perpendicular to the ipsilateral thyroid lamina, as described in our previous experiences [13].

The four patterns identified were therefore as follows:
1)Supraglottic carcinoma encasing the arytenoid from above with its fixation due to weight effect, without direct involvement of the CAU (Figure 2):
–The tumor infiltrates the superior paraglottic space (SPGS) and the thyro-arytenoid muscle (TAM) without direct involvement of the crico-arytenoid unit (CAU), with evaluation of the following radiological parameters: superior paraglottic space (SPGS), TAM, inferior paraglottic space (IPGS), crico-arytenoid joint (CAJ) and cricoid plate.
2)Glottic carcinoma involving the IPGS and extending toward the CAU, medially contained by the conus elasticus and laterally by the thyroid wing (Figure 3):
–The tumor extends in part to the SPGS becoming transglottic. –It involves TAM, IPGS, thyro-arytenoid space and/or posterior crico-thyroid space and blocks the arytenoid laterally corresponding to the lateral crico-arytenoid muscle; the tumor involves or invades the arytenoid but not the cricoid (that can present signs of sclerosis anyway), and there is the strong doubt about CAJ involvement, even if in part.–The eventual infraglottic extension, evaluated exactly at the free edge of the TVC midline, is ≤10 mm.
3)Glottic-subglottic carcinoma invading the CAJ, and wrapping around it, because the tumor can have type II progression together with CAJ invasion through the conus elasticus with medial to lateral progression (Figure 4):
–The tumor massively involves the IPGS, the thyro-arytenoid space and/or posterior crico-thyroid space and blocks the arytenoid laterally corresponding to the lateral crico-arytenoid muscle, it involves or invades the arytenoid and often the cricoid. Cricoid and CAJ involvement are frequent, through the conus elasticus.–Infraglottic extension >10 mm, evaluated exactly at the free edge of the TVC midline, with progression toward the posterior commissure.–It tends to minimally extend upwards toward the SPGS.
4)Transglottic and infraglottic carcinoma with massive involvement of the CAU and of the posterior crico-arytenoid muscle, reaching the hypopharyngeal submucosa (Figure 5):
–The tumor massively infiltrates the SPGS and IPGS, the thyro-arytenoid space and/or posterior crico-thyroid space, it blocks the arytenoid laterally corresponding to the lateral crico-arytenoid muscle, extending downwards medially into the infraglottic site and puncturing the conus elasticus.–Infraglottic extension >10 mm, evaluated exactly at the free edge of the TVC midline, with progression toward the posterior commissure.–The tumor invades the arytenoid and, very often, the cricoid, involving the posterior crico-arytenoid and the interarytenoid musculature, reaching the hypopharyngeal submucosa.–Possible involvement of the contralateral CAU.



### 2.4. Pathologic Assessment

Considering that two out of three patients belonging to pattern 1 were subjected to non-surgical treatment, the study focused only on patterns 2–4. Twenty-six specimens, nine OPHL and 17 TL, were evaluated by a pathologist experienced in head and neck cross-sectional study, and subsequently blind reviewed by two pathologists to assess the concordance with the radiological pattern of arytenoid fixation.

The samples have been preserved in fixative solution (buffered formalin at 10%) for 16 hours; then the material was placed in descaling solution (3M formic acid) for about 24 h.

After descaling treatment, each sample was examined entirely with serial horizontal and sagittal sections, about 5 mm thickness, following the method described by Nakayama et al. [16].

To evaluate the degree of subglottic extension, the distance between the glottic free edge and the inferior edge of the tumor was assessed macroscopically and microscopically at the true vocal cord midline.

### 2.5. Treatments

After informed consent had been obtained, surgery was performed accordingly with the AIOM guidelines [14]. The different types of treatment performed are reported in Table 2. 

Nine patients underwent OPHL types II–III according to the European Laryngological Society Classification [17], 17 patients underwent total laryngectomy (TL).

Neck dissection (ND), graded according to the American Academy of Otolaryngology—Head and Neck Surgery Foundation Classification [18], was performed in 25 patients (96.2%), and was unilateral in 8 (30.8%) and bilateral in 17 (65.4%). ND was performed electively (ND levels II–IV) in 15 cN0 patients (57.7%) and for curative purposes in 10 cN > 0 (38.5%). In 16 patients (61.5%), level VI or unilateral paratracheal lymph node clearance was added. In one elderly patient (90 yo, cN0), no ND was performed.

### 2.6. Adjuvant Treatments

Fifteen patients (57.7%) were subjected to adjuvant radiotherapy. The indications were: pN+ > 1 (*n* = 5 patients), gross extralaryngeal extension (*n* = 15 patients), and positive margins elsewhere (*n* = 1). A large volume encompassing the primary site and all draining lymph nodes were irradiated with a dose of up to 54 Gy. Regions at higher risk for malignant dissemination received a 12 Gy boost (total, 66 Gy; range, 62–68 Gy).

Furthermore, chemotherapy was added in 10 patients who received 100 mg/m^2^ of cisplatin on days 1, 22, and 43, concomitantly with radiotherapy because of a higher risk of local recurrence (four with extracapsular spread, and six with more extended pT4a showing close margins toward pre-laryngeal tissues and perineural invasion) [19].

### 2.7. Statistical Methods

The accuracy of the imaging diagnosing the studied disease has been evaluated with 2×2 contingency tables, considering pathology as the gold standard. Clinical, endoscopic, and radiologic follow-up was performed for a mean of 3 years (range, 19 months–4.4 years). Overall survival (OS), locoregional control (LRC) and laryngo-esophageal dysfunction-free survival (LEDFS) [20] were assessed by Kaplan–Meier curves. Log-rank (LR) and, for early events, Gehan-Breslow-Wilcoxon (GBW) tests were used to compare Kaplan–Meier estimates among the different patterns. The end points considered were: the date of death (OS); the date of the first locoregional recurrence (LRC); the date of salvage total laryngectomy or the date of tracheostomy and/or percutaneous endoscopic gastrostomy (PEG) for functional reasons or the date of death (LEDFS). The association of prognostic factors for recurrence and subcategories was evaluated by odds ratio, meanwhile the corresponding incidences were compared by chi-squared (χ^2^) test. All analyses were performed with GraphPad Prism version 6.0e (GraphPad Software, San Diego, CA, USA), with *p* < 0.05 as the threshold for statistical significance.

## 3. Results

The correlation between radiologic and pathologic findings for arytenoid fixation patterns 2–4 is reported in Table 3. 

SPGS, IPGS, ARY, CAJ, cricoid plate, subglottic extension >10 mm and contralateral CAU (any of the following: ARY, CAJ, cricoid plate) involvement were analyzed according to the aforementioned pattern.

The overall CAJ involvement was 65.5% (40.0% in pattern 2, 76.5% in pattern 3 and 100% in pattern 4). The overall subglottic extension >10 mm at the midline of the TVC was 62.1% (20.0% in pattern 2, 82.4% in pattern 3 and 75.0% in pattern 4). The cricoid plate was involved by the tumor in 10.3% (0.0% in pattern 2, 11.8% in pattern 3 and 25.0% in pattern 4). Overall contralateral CAU involvement was 3.44% (0.0% in pattern 2, 0.0% in pattern 3 and 25% in pattern 4).

Among the 21 cases that were studied with conventional MRI (T2-weighted images acquired on the axial and coronal planes and T1-weighted images acquired on the axial plane), a peritumoral inflammation determining a drop out of specificity in the evaluation of PGS involvement was detected in 47.6% of cases. In particular, in the nine cases who underwent OPHL, the more advanced magnetic resonance techniques (diffusion weighted imaging—DWI) allowed better discrimination of tumor from inflammation. 

In practice, the improvement in work-up resulted in a modular approach (informed consent obtained from the patient to shift to more extended OPHL or even to total laryngectomy) in carrying out the OPHL and an immediate check of radicality by frozen section. A positive margin (negative at frozen section but positive at the definitive histopathologic examination) was present in only one case (11.1%) in pyriform sinus sub-mucosa due to SPGS involvement. Better specificity is of great importance in the decision to carry out an OPHL in tumors recurring after radiotherapy (Figure 6). At pathological classification, the cohort were stratified as 4 pT3 and 1 pT4a in pattern 2, 6 pT3 and 11 pT4a in pattern 3, and 4 pT4a in pattern 4.

In total, eight (27.6%) patients developed recurrences: five (62.5%) regional, one (12.5%) regional and distant, whereas two (25%) developed distant metastasis only. 

After 3 years, LEDFS was 66.7% of cases in pattern 1, 80% in pattern 2, 7.8% in pattern 3 and 0.0% in pattern 4.

The 3-year LEDFS estimates of the above-mentioned oncologic outcomes for each subcategory are reported in Table 4. 

LEDFS was significantly higher (*p* < 0.001, with LR and GBW tests) in patients affected by tumors classified in pattern 2 (75%) when compared with those belonging to patterns 3 and 4, (6.3%) (Figure 7).

## 4. Discussion

In the head and neck district, the larynx is one of the most paradigmatic sites, because of its communicative and functional valence, as voice, swallowing and breathing. In the absence of precise prognostic clinic-biological nomograms that can influence the therapeutic choice, the major possibilities of success in conservative treatments depend on the work-up accuracy on T and N; in particular, the choice of conservative treatment, by OPHL or CRT, rather than extirpative surgery, is governed by the precision of T and N staging. Regarding the latter, one must also consider the relatively low incidence of clinically positive lymph nodes in intermediate-advanced cases: pN > 0 in 4.4% of pT2, and pN > 0 in 12.8% of pT3–pT4a [21,22].

Assuming that total laryngectomy is the safest treatment for locally advanced laryngeal cancer, sometimes OPHL can be offered to patients as a valid alternative for preserving part of the larynx and its functions, and also in intermediate-advanced tumors, thus avoiding the physical and psychological impact of permanent tracheostomy. 

Recently, our group identified and adopted simple clinic-radiological criteria in a series of 479 cases managed with OPHL (matching the transgression or not of a vertical virtual plane tangential to the arytenoid vocal process and perpendicular to the ipsilateral thyroid lamina, and the functional data on arytenoid fixation), in order to identify homogenous subcategories of cT3 and cT4 LC treatable by OPHL with the best possibilities of success [13]. For both cT3 and cT4 glottic and supraglottic tumors, the most important negative prognostic factor seems to be IPGS involvement, in its posterior part, generally associated with homolateral arytenoid impaired mobility or fixation. This simple criterion of anterior vs posterior anatomo-functional compartmentalization of the larynx proved to be a useful additional parameter in preoperative planning.

In fact, in anterior T3 tumors with normal arytenoid/chordal mobility (pattern 1) and in anterior T4a tumors too (pattern 3), OPHL allows quite favorable oncologic outcomes that can be compared only with total laryngectomy outcomes: here the difference is in terms of best quality of life and preservation of laryngeal function.

On the other hand, posterior lesions are more difficult to manage by conservative treatment and represent, finally, a harder clinical scenario. Even using OPHLs, T3 posterior tumors, with fixed vocal cord and arytenoid (pattern 2), showed significantly worse oncologic outcomes (OS *p* < 0.001, DSS *p* < 0.05, and DFS *p* < 0.001) when compared with anterior tumors.

Considering that even in more posterior tumors with fixed arytenoid, we can achieve complete surgical radicality by more extended horizontal partial laryngectomies, in this study, we analyzed 29 consecutive cases of laryngeal cancer classified as cT3 and cT4a, and treated in our Institute, with the aim of better understanding the real value of the fixed arytenoid parameters in terms of risk factors for managing the disease by OPHL and to determine whether there were other factors which might be considered for better definition of the tumor.

Historically, in fact, arytenoid fixation has been considered a negative prognostic factor and a contraindication for partial laryngectomy, as a sure sign of CAU invasion, involving the cricoid plate, arytenoid, and musculature as well as directly invading the recurrent nerve [3,5,8,9,10].

Since 2006, with the introduction of type III OPHL [23], that resects the CAU, and with the introduction of the OPHL classification since 2014 [17], the scenario has changed, because surgical equipment has advanced. This has resulted in a greater attention to detail which modern imaging can offer.

In our opinion, the results of this study are encouraging in this sense, allowing us to identify 4 different types of arytenoid fixation, graded by different patterns of tumor extension, which correspond to different types of treatment, conservative and/or extirpative.

In pattern 1, arytenoid fixation is linked to the weight effect, because it concerns supraglottic “bulky” tumors with SPGS and IPGS involvement, which do not generally involve the crico-arytenoid joint.

In pattern 2, arytenoid fixation is linked to tumor evolution toward the thyro-arytenoid and crico-thyroid spaces and, finally, toward the CAJ and the lateral crico-arytenoid muscle.

In 1996, Laccourreye stated that neither CT scan nor MRI were able to distinguish the cricoarytenoid muscle from cricoarytenoid joint invasion [24]. As demonstrated by Maroldi et al., modern MRI with surface coils and thin slices (3 mm) has several advantages compared with multislice CT, especially in determining tumor extension toward the IPGS and the CAJ [25]. In this evolution, the neoplasm is contained medially by the conus elasticus and laterally by thyroid cartilage; these two structures tend to channel the neoplastic evolution toward an extralaryngeal progression through the crico-thyroid space, determining a funnel effect. With regard to infraglottic evolution, the chosen cut-off, meaning a subglottic extension <10 mm from the midline of the free edge of the TVC, ensures that the tumor does not massively involve these anatomic structures yet, but it is just at the beginning of subglottic evolution (Figure 8). 

In our series, five cases are represented in this subcategory, all of them treated with partial laryngectomy (three OPHL type IIa + ARY, two OPHL type IIIa + CAU). In all these patients a complete resection was achieved, with free margins; they are all alive and free from disease and they all have preserved laryngeal function.

Specimen analysis showed CAJ involvement in two cases (25.0%) with imaging sensibility and specificity of 100%; and absence of cricoid invasion (0.0%), with imaging sensibility and specificity of 100%. Very schematically, it can be stated that, in the case of a glottic tumor, even with transglottic evolution and fixed arytenoid but with subglottic extension ≤10 mm, the risk of involvement of one or more anatomic elements of the CAU is low and, at most, focusing on the tumor itself. 

The eventual proposable partial laryngectomy is OPHL type IIa + ARY, possibly extending to the cricoid rostrum, as described by Biller and Lawson in 1986 in an enlarged hemilaryngectomy, or a safer, in terms of resection, OPHL type IIIa + CAU [23,26].

Today, OPHL type IIa + ARY extending to the cricoid rostrum is made easier by the possibility of using a very precise cutting tool, such as the CO_2_ fiber laser, allowing us to extend an OPHL type IIa downward to include every anatomic structure that composes the CAU and finally reaching the same surgical radicality as a supratracheal laryngectomy, without weakening the rigid annular structure of the cricoid too much, and avoiding annoying dysfunctional respiratory sequelae, described following a supratracheal laryngectomy. Figure 9 shows the resection scheme for OPHL type IIa + ARY extending to the cricoid rostrum.

The risk of treating a tumor in this subcategory with a partial laryngectomy is represented by the possible extralaryngeal extension (only one out five patients were pT4a = 20%) through the crico-thyroid space with risk of involvement of the thyroid lobe; therefore, in the work-up, it is important to pay close attention to radiological evaluation of this aspect, normally accompanied by an erosion of the inferior aspect of the thyroid cartilage and a suspected involvement of the adjacent superior border of the cricoid.

In this pattern, it is extremely important to discriminate between tumor and peritumoral inflammation in the PGS. Therefore, in agreement with Maroldi et al. [5], we consider the work-up by MRI and DWI sequences superior to multislice CT in the study of patients undergoing OPHL. This result is of particular importance when this type of surgery is carried out on more advanced cases, as resection margins are inevitably reduced to a few millimeters [16].

In subcategory III, arytenoid fixation is linked not only to neoplastic evolution toward the thyro-arytenoid and crico-arytenoid spaces, with a tendency for invasion of the CAJ and the lateral crico-arytenoid muscle, but also to the evident subglottic evolution, with the possibility that the tumor involves the CAJ and the cricoid medially to laterally, through the conus elasticus. In this pattern, the potentiality to spread toward extra-laryngeal sites, laterally through the crico-thyroid space and posteriorly through the cricoid, is a further concern (11 out 17 patients resulted pT4a = 64.7%) [3,9].

In our series, 17 cases are represented in this subcategory, 14 managed with total laryngectomy and three with OPHL type IIIa + CAU. In 100% of TL cases, a complete resection was achieved with negative margins; 76.9% are alive and free from disease. In all three OPHL type IIIa + CAU cases, complete resection was achieved with negative margins; they are all alive and free from disease, two out of three with laryngeal function preserved (one complete laryngectomy for severe laryngeal stenosis).

Specimen analysis showed CAJ involvement in 13 cases (76.5%), with imaging sensibility of 84.6% and specificity of 75.0%; and a cricoid invasion in two (11.8%), with imaging sensibility of 50.0% and specificity of 80.0%. 

Schematically, it can be stated that, in the case of a glottic-subglottic tumor, even with a modest transglottic evolution, but with subglottic extension >10 mm, the risk of involvement of one or more anatomic elements of the CAU is very high, and equating this pattern to a T4a.

In this case, the unique “rational” treatment is represented by total laryngectomy, which produces a complete and “en bloc” removal of all laryngeal musculature, intrinsic and extrinsic. In the case of categorical refusal of total laryngectomy by the patient, a conservative treatment by concurrent chemoradiotherapy (CCRT), IC+RT or partial laryngectomy can be applied. The only proposable partial laryngectomy is an OPHL type IIIa + CAU, whose infero-posterior resection limits are the first tracheal ring and the retro-cricoid mucosa.

In this case too, the possibility to use a very precise cutting tool, such as the CO_2_ fiber laser, used by the free hand in open surgery, allows all of the anatomic structures composing the CAU to be understood, but, in particular, demands great precision in the dissection and separation of very narrow anatomic layers, in order to check, intraoperatively, eventual extra-laryngeal extensions determining the necessity to convert the partial operation into an extirpative one.

Similarly, an organ preservation protocol with CCRT or IC + RT can be applied in the hope that suspected cartilaginous involvement is actually a false positive of the imaging. In this situation, in order to increase sensibility and specificity in the preoperative definition of the clinical case, it can be proposed to execute both CT, which reduces acquisition time and improves image quality, and MRI, with greater definition of cartilage.

In this subcategory, the weakest point is represented by frequent extralaryngeal extension toward the crico-thyroid space with involvement of the thyroid lobe and, posteriorly, through the thyro-arytenoid space and the cricoid; therefore, close attention must be paid to the radiologic evaluation of these aspects not only in the work-up, but also during the eventual partial laryngectomy, that must be carefully “controlled” intraoperatively with precise frozen exams. In this case too, the reduced thermic damage of the CO_2_ fiber laser is very useful to the pathologist.

Lastly, in subcategory IV, arytenoid fixation is linked to the sum of different types of evolution, with the tumor reaching the hypopharyngeal submucosa and involving the contralateral CAU through a progression by way of the posterior or anterior commissure. In this evolution, the tumor is, de facto, always a T4a. Four out four patients were pT4a at pathological staging.

In our series, four cases are represented in this subcategory, three managed with total laryngectomy and one with OPHL type IIIa + ARY. In this last case, despite the frozen exam negativity, specimen analysis showed a positive posterior margin. Despite adjuvant CRT, the patient suffered a relapse after 8 months and died from locoregional progression of the disease.

Specimen analysis showed CAJ involvement in 100% of cases, with imaging sensibility and specificity of 100% (4/4); and cricoid invasion in 1/4 cases (25.0%), with imaging sensibility of 100% and specificity of 33.3%.

In this pattern, the unique proposable treatment is total laryngectomy, with complete “en bloc” removal of all intrinsic and extrinsic laryngeal musculature and that must extend to the hypopharyngeal mucosa and also to the thyroid lobe.

The weak point of each surgical treatment of a tumor belonging to this subcategory is the frequent involvement of hypopharyngeal mucosa, which must always be suspected during the clinical-radiological evaluation and evaluated intraoperatively with frozen sections of the mucosal resection margin.

## 5. Conclusions

At the end of our studies on the simplification and refinement of selection of tumors manageable with OPHLs, always bearing in mind the value of patient related indications/contraindications, we can summarize our results as follows: a) the larynx compartmentalization criterion into an anterior and a posterior part, according to arytenoid mobility/fixation, is a simple parameter that allows identification of cases safely manageable with OPHL, i.e., those involving just the anterior compartment with mobile arytenoid; b) this parameter can be applied to cT3 tumors as well as, with great care, anterior cT4a tumors; c) tumors involving the posterior compartment of the larynx causing arytenoid fixation must be studied carefully during work-up, because of the frequent involvement of the laryngeal framework and because of the effect of the extralaryngeal extension; d) in this tumoral subcategory too, a small subset of cases still safely manageable with a conservative (surgical or non-surgical) approach can be identified (25% in our experience); these cases are those where arytenoid fixation occurs when the tumor does not extend to the subglottis (pattern 2); e) in cases where arytenoid fixation is related to a subglottic extension >10 mm (pattern 3) or to massive posterior CAU invasion (pattern 4), the tumor clearly acts as a cT4a, so the only sure and rational treatment is total laryngectomy, considering also some negative events (positive margins, level VI pN+, extracapsular spread (ECS)), that could determine more frequent recurrences; f) in the case of categorical refusal of the patient to undergo total laryngectomy, after informing him/her about the risk of giving priority to functional outcome rather than to oncologic outcome, in some very selected cases (pattern 3), it is still possible to resort to OPHL type IIIa. This option, even though not optimal, should be equally considered with CCRT in the drafting of guidelines.

## Figures and Tables

**Figure 1 cancers-11-00360-f001:**
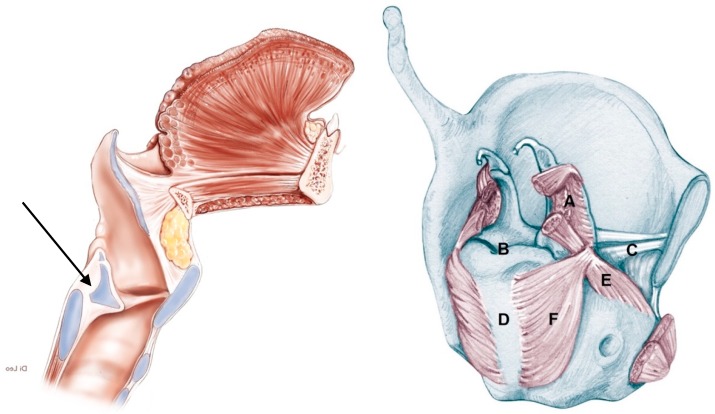
General view of the larynx in the sagittal plane (**left**) and a particular view of the crico-arytenoid unit (CAU) (**right**)—Mean percentages of neoplastic involvement of the CAU’s components in the case of arytenoid fixation: (A) Inter-arytenoid muscle: 57.8%; (B) Crico-arytenoid joint: 40.1%; (C) Thyro-arytenoid muscle: 71.8%; (D) Cricoid plate: 73.3%; (E) Lateral crico-arytenoid muscle: 73.3%; (F) Posterior crico-arytenoid muscle: 30.0%.

**Figure 2 cancers-11-00360-f002:**
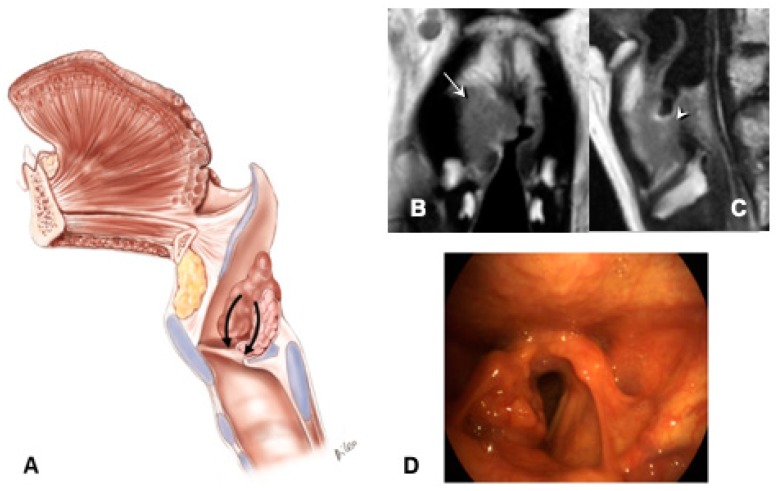
Pattern 1: Supraglottic carcinoma encasing the arytenoid from above with fixation due to weight effect, without direct invasion of the CAU: (**A**) sagittal scheme. (**B**) Coronal T2-weighted MR image shows a supraglottic tumor that involves the superior paraglottic space on the right side (arrow) and invades the glottis. (**C**) Sagittal T2-weighted image demonstrates that the lesion encases the right arytenoid that appears diffusely hypointense because of sclerosis suspicious for invasion (arrowhead), without clear evidence of CAU involvement. (**D**) Endoscopic view.

**Figure 3 cancers-11-00360-f003:**
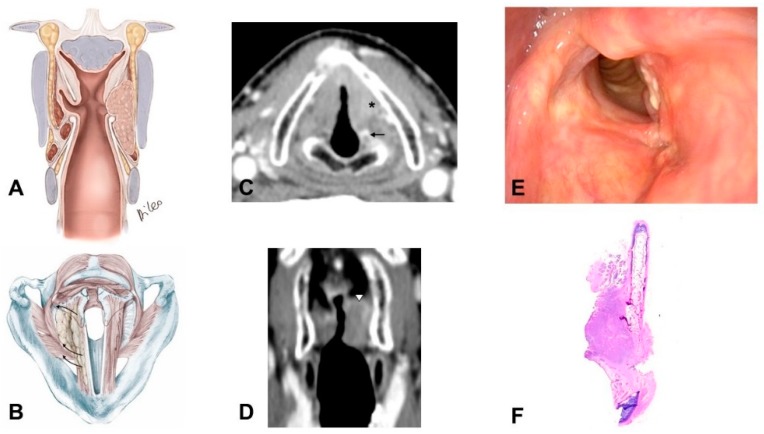
Pattern 2: Glottic carcinoma invading the inferior paraglottic space and extending down to the CAU, limited medially by the conus elasticus and laterally by thyroid cartilage: (**A**) Coronal scheme; (**B**) Axial scheme; (**C**) Axial CE-CT at vocal process level: enhanced neoplastic tissue involves the left true vocal cord and anterior commissure; the inferior paraglottic space is obliterated (asterisk). Posteriorly, the lesion is close to the left arytenoid cartilage which shows slight sclerosis (arrow); (**D**) CE-CT reconstruction on the coronal plane: the lesion partially invades the left superior paraglottic space becoming a so-called transglottic tumor (arrowhead). Subglottic neoplastic spread > 10 mm is not present; (**E**) Endoscopic view; (**F**) Whole organ section in sagittal view: tumor clearly invades the inner cortex of the thyroid cartilage reaching the inferior aspect of the latter.

**Figure 4 cancers-11-00360-f004:**
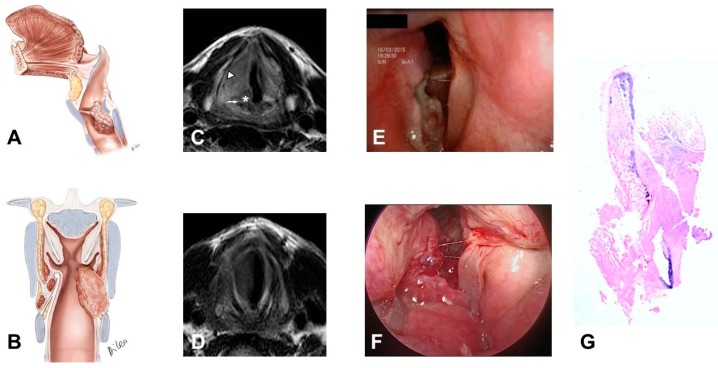
Pattern 3: Glottic-subglottic carcinoma invading and enveloping the crico-arytenoid joint. The tumor shows medial to lateral growth through the conus elasticus: (**A**) Sagittal scheme; (**B**) Coronal scheme; (**C**) Axial T2-weighted MR image obtained at vocal process level demonstrates a right-sided true vocal cord tumor involving the inferior paraglottic space (arrowhead). The right arytenoid cartilage shows intermediate signal intensity alteration (asterisk) indicative of invasion; crico-arytenoid unit involvement is observed (arrow); (**D**) Axial T2-weighted sequence shows the subglottic spread of the neoplastic mass >10 mm; (**E**) Endoscopy at vocal cord level; (**F**) Endoscopy at subglottic level; (**G**) Whole organ section in sagittal view: tumor clearly invades the inferior aspect of the thyroid cartilage and the superior aspect of the cricoid with early extra-laryngeal spread.

**Figure 5 cancers-11-00360-f005:**
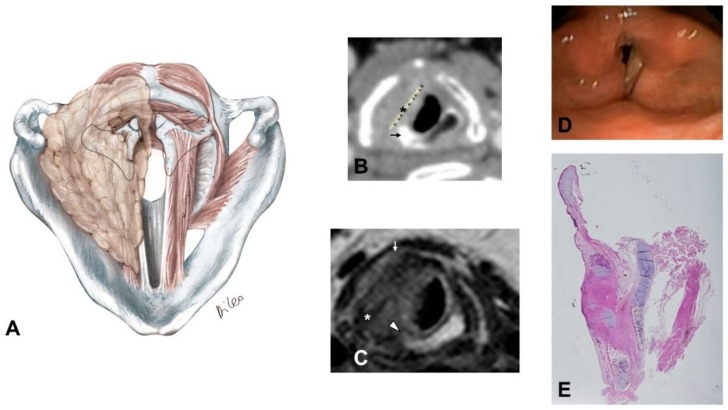
Pattern 4: Transglottic-hypoglottic carcinoma with massive invasion of the crico-arytenoid unit and involvement of the posterior crico-arytenoid muscle, reaching the hypopharyngeal submucosa: (**A**) Axial scheme; (**B**) Axial CE-CT image at the level of the cricoid lamina shows a right-side neoplastic mass with homogeneous enhancement (asterisk). The right cricoid cartilage demonstrates sclerosis suspicious for neoplastic invasion (arrow); (**C**) Axial T2-weighted MR image at the same level: right cricoid cartilage shows signal alteration (arrowhead) similar to intermediate tumor signal intensity, indicative of invasion. The lesion extends to the right crico-thyroid space which appears enlarged (asterisk). Thyroid cartilage is invaded and extra-laryngeal neoplastic spread is observed (arrow); (**D**) Endoscopic view (**E**) Whole organ section in sagittal view: massive involvement of the cricoid plate.

**Figure 6 cancers-11-00360-f006:**
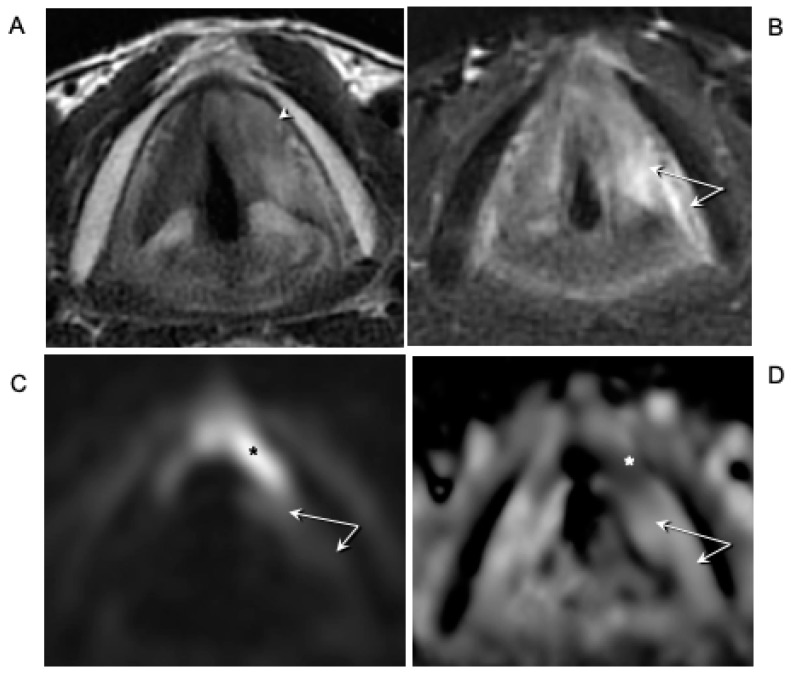
Left glottic radio-recurrent squamous carcinoma recently treated by OPHL type IIa + left Ary. The endoscopic findings showed a fixed vocal cord with marked impairment of mobility of the left arytenoid. The pathology confirmed the absence of posterior paraglottic space involvement. (**A**) Axial T2-weighted MR image demonstrates an intermediate signal intensity neoplastic tissue involving the anterior third of the left true vocal cord that extends to the anterior commissure with initial spread on the right side; invasion of the left inferior paraglottic space is present (arrowhead). (**B**) Axial T2-weighted image with fat suppression shows more hyperintense signal alterations in the posterior aspect of the left vocal cord and the posterior paraglottic space (arrows). (**C**) On DWI images (b-value 1000 s/mm^2^), the tumor demonstrates hyperintensity (black asterisk) that corresponds to hypointensity (white asterisk), on the ADC map (**D**) reflecting diffusion restriction; no abnormal alterations on DWI or on the ADC map are observed in the posterior aspect of the left vocal cord and in the posterior paraglottic space indicating peritumoral edema (arrows).

**Figure 7 cancers-11-00360-f007:**
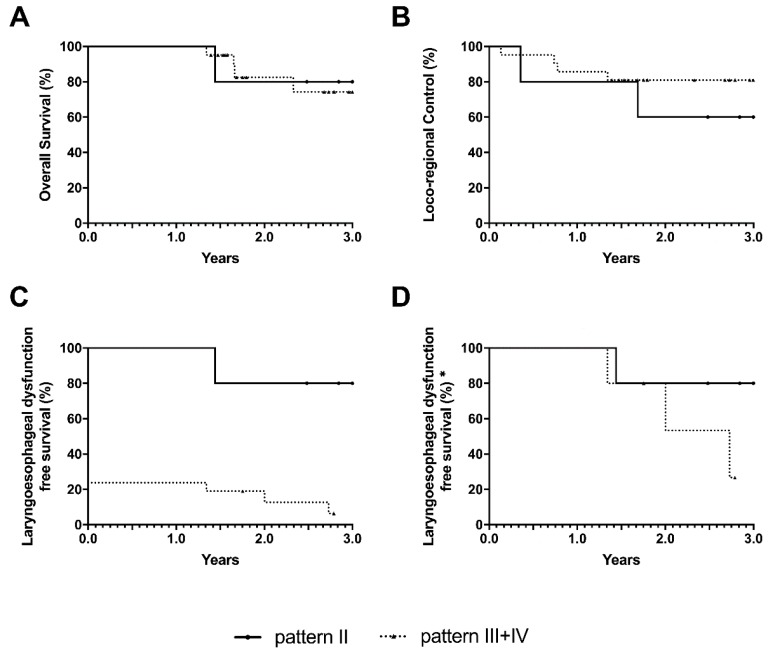
Oncologic outcomes at 3 years for pattern 2 compared with those belonging to pattern 3–4: (**A**) Overall survival; (**B**) Locoregional control; (**C**) Laryngo-esophageal dysfunction-free survival; (**D**) Laryngo-esophageal dysfunction-free survival (* without considering patients undergoing up-front total laryngectomy).

**Figure 8 cancers-11-00360-f008:**
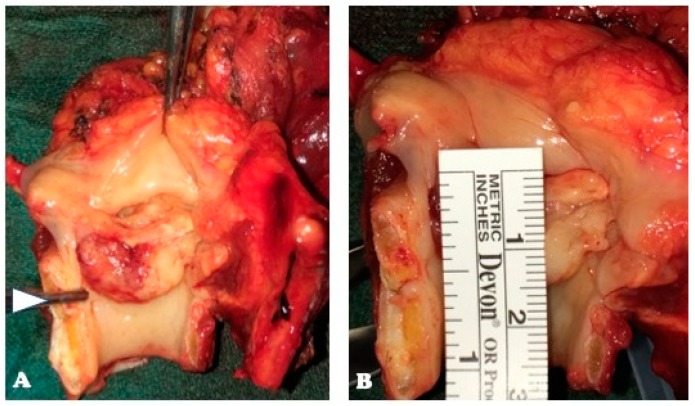
Specimen of OPHL type IIIa + CAU: glottic -subglottic LC showing infraglottic extension >10 mm: (**A**) Tumor shows an evident subglottic extension, clearly reaching above the level of the cricoid plate (white arrow); (**B**) Tumor spreads infraglottically 15 mm from the free edge of the TVC midline.

**Figure 9 cancers-11-00360-f009:**
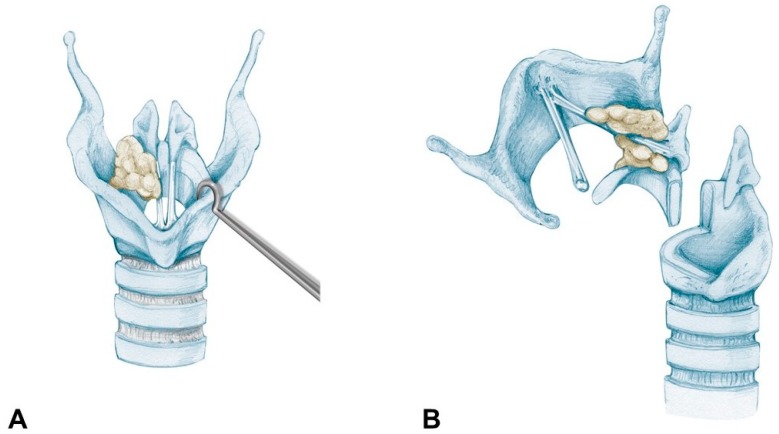
OPHL type IIa + (right) ARY extending above the cricoid at risk of focal involvement of the cricoid plate: (**A**) Before the resection: glottic carcinoma invading the posterior paraglottic space and extending down to the CAU; (**B**) The OPHL IIa + right ARY resection is extended to a part of the cricoid plate corresponding to the superficial extension of the tumor.

**Table 1 cancers-11-00360-t001:** Epidemiologic and clinical characteristics of patients treated in the present series (*N* = 29); clinical TN classification (cTN) because patients treated only by chemoradiotherapy (CRT).

Parameters		*N* of Patients (%)	
**Age**	Mean ± standard deviation	65.4 ± 9.4	
	Range	50–90	
**Gender**	Male	23 (79.3%)	
	Female	6 (20.7%)	
**Karnofsky**	100	24 (82.8%)	
	90	1 (3.4%)	
	80	4 (13.8%)	
**Arytenoid mobility**	Fixed	29 (100%)	
**cTN**		**Glottic**	**Supraglottic**
cT3	N0	8 (27.6%)	2 (6.9%)
	N1	2 (6.9%)	0 (0.0%)
	N2	0 (0.0%)	0 (0.0%)
cT4a	N0	8 (27.6%)	1(3.5%)
	N1	2 (6.9%)	2 (6.9%)
	N2	4 (13.8%)	0 (0.0%)
Total		24 (82.8%)	5 (17.2%)

**Table 2 cancers-11-00360-t002:** Treatments performed in the 29 patients included in this study.

Type of Treatment		Pattern 1 (*N* = 3)	Pattern 2 (*N* = 5)	Pattern 3 (*N* = 17)	Pattern 4 (*N* = 4)	Total
**TL**		1 (33.3%)	0 (0.0%)	14 (82.4%)	3 (75.0%)	18 (62.0%)
OPHL	IIa + ARY, *n* (%)	0 (0.0%)	3 (60.0%)	0 (0.0%)	0 (0.0%)	3 (10.3%)
	IIb + ARY, *n* (%)	0 (0.0%)	0 (0.0%)	0 (0.0%)	0 (0.0%)	0 (0.0%)
	IIIa + CAU, *n* (%)	0 (0.0%)	1 (20.0%)	3 (17.6%)	1 * (25.0%)	5 (17.2%)
	IIIb + CAU, *n* (%)	0 (0.0%)	1 (20.0%)	0 (0.0%)	0 (0.0%)	1 (3.5%)
IC + RT		2 (66.7%)	0 (0.0%)	0 (0.0%)	0 (0.0%)	2 (7.0%)

TL: total laryngectomy; IIa: supracricoid partial laryngectomy with cricohyoidoepiglottopexy; IIb: supracricoid partial laryngectomy with cricohyoidopexy; IIIa: supratracheal partial laryngectomy with tracheohyoidoepiglottopexy; IIIb: supratracheal partial laryngectomy with tracheohyoidopexy; IC + RT: induction chemotherapy + radiotherapy; ARY: removal of one arytenoid; CAU: removal of one cricoarytenoid unit; *: positive margins.

**Table 3 cancers-11-00360-t003:** Comparison between imaging and pathological evaluation results.

Invaded Structures	Pattern 2 (*N* = 5)		Pattern 3 (*N* = 17)		Pattern 4 (*N* = 4)	
	Imaging	Pathology	Imaging	Pathology	Imaging	Pathology
**SPGS**	3	2	17	13	3	2
**IPGS**	5	5	17	17	4	4
**ARY**	4	4	16	13	4	4
**CAJ**	2	2	12	13	4	4
**Cricoid plate**	0	0	4	2	3	1
**Subglottic extension >10mm**	0	1	17	14	4	3
**Contralateral CAU**	0	0	0	0	1	1

SPGS: superior paraglottic space; IPGS: inferior paraglottic space; ARY: arytenoid CAJ: crico-arytenoid joint; CAU: crico-arytenoid unit.

**Table 4 cancers-11-00360-t004:** Three-year LEDFS estimates.

End point	Pattern			
	2	3	4	3 + 4
**OS**	80.0%	75.2%	75.0%	74.3%
**LRC**	60.0%	88.2%	50.0%	81.0%
**LEDFS**	80.0%	33.3%	0.0%	6.3%
**LEDFS ***	80.0%	33.3%	0.0%	26.7%

OS: overall survival; LRC: locoregional control; LEDFS: laryngo-esophageal dysfunction-free survival; LEDFS *: laryngo-esophageal dysfunction-free survival without considering patients undergoing up-front total laryngectomy.

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
