# Peer review of "Arytenoid Fixation in Laryngeal Cancer: Radiological Pictures and Clinical Correlations with Respect to Conservative Treatments"

_cancers, 2019, doi:10.3390/cancers11030360_

Round 1
Reviewer 1 Report
The Authors describe an extremely selected series of laryngeal cancer patients presenting with arytenoid fixation. Figures (drawings, imaging, and histopathologic examination) provide a novel and precise representation of such tumors (I would consider increasing their size).
This is an interesting manuscript with an extremely up-to-date and discussed topic. However, some changes are needed before considering the manuscript for publication.
The possibility of arytenoid fixation due to peritumoral inflammation has not been considered. I would better discuss this aspect and specify its impact on your series.
Two patients in this series were treated by induction chemotherapy and radiotherapy. For this reason, postoperative histopathology is not available and correlations with radiology have been performed by “deduction”. In this view, there is no hard data supporting your conclusions. Considering that 2 out of 3 patients in Pattern 1 received this kind of treatment, i would consider excluding Pattern 1 from the study.
As stated by the Authors, the study “had as unique objective the attempt to identify some homogeneous subcategory in the large group of lesions determining arytenoid fixation, in order to understand which cases could be liable of conservative laryngeal treatment with sufficient security.” In this view, I would better describe and characterize survival results and patterns of recurrence. The subsequent discussion should be more focused on the development of a diagnostic and therapeutic algorithm for the different lesions presenting with arytenoid fixation.
Author Response
The Authors describe an extremely selected series of laryngeal cancer patients presenting with arytenoid fixation. Figures (drawings, imaging, and histopathologic examination) provide a novel and precise representation of such tumors (I would consider increasing their size).
Author’s response: We thank the Reviewer for these statements. As suggested, where possible, we have increased the size of the figures.
This is an interesting manuscript with an extremely up-to-date and discussed topic. However, some changes are needed before considering the manuscript for publication.
The possibility of arytenoid fixation due to peritumoral inflammation has not been considered. I would better discuss this aspect and specify its impact on your series.
Author’s response: The manuscript has been reconsidered in light of this recommendation. The peritumoral inflammation was discussed from the eleventh row of the “Results” section.
Two patients in this series were treated by induction chemotherapy and radiotherapy. For this reason, postoperative histopathology is not available and correlations with radiology have been performed by “deduction”. In this view, there is no hard data supporting your conclusions. Considering that 2 out of 3 patients in Pattern 1 received this kind of treatment, i would consider excluding Pattern 1 from the study.
Author’s response: In the new version of the manuscript, Pattern 1 was retained to describe the clinical-radiological features, but excluded from the subsequent analyses.
As stated by the Authors, the study “had as unique objective the attempt to identify some homogeneous subcategory in the large group of lesions determining arytenoid fixation, in order to understand which cases could be liable of conservative laryngeal treatment with sufficient security.” In this view, I would better describe and characterize survival results and patterns of recurrence. The subsequent discussion should be more focused on the development of a diagnostic and therapeutic algorithm for the different lesions presenting with arytenoid fixation
Author’s response: We agree with the Reviewer. A further analysis of survival results has been performed and data regarding locoregional control has been added to Table 4. The analysis of disease-specific survival revealed its superimposition with overall survival, as all exitus were caused by the disease. For this reason, DSS was not included. Finally, no local recurrences were detected in our series.
Reviewer 2 Report
Dear Authors,
Congratulations on the successful completion and presentation of results of the study titled 'Arytenoid Fixation in Laryngeal Cancer: Radiological Pictures and Clinical Correlations with Respect to Conservative Treatments'. The presentation of results in the manuscript is very good, but there are a few concerns to be addressed before the manuscript becomes suitable for publication in Cancers.
1. Introduction: The figure in the introduction section could have another image which can show the position of the crico-arytenoid unit in the human body. From which the CAU could be shown as a magnified version.
2. The grammar and the construction of some sentences needs to be corrected in the introduction section. For ex: "This retrospective mono-institutional study, conducted on a consecutive series of 29 patients affected by T3-T4a untreated LC, had as unique objective the attempt to identify some homogeneous subcategory in the large group of lesions determining arytenoid fixation, in order to understand which cases could be liable of conservative laryngeal treatment with sufficient security."
3. Figure 2,3,4 and 5: Labeling the figure with major parts will make the image more readable and easy to understand. Without which the images are not identifiable.
4. The manuscript has various sections where English grammar has to be corrected to increase the quality of the content. For example: "About the infraglottic evolution, the chosen cut-off, meaning a subglottic extension < 10 mm from at the midline of the free edge of TVC ensures the fact that the tumor does not interested massively these anatomic structures yet". These are some examples of poor English in the manuscript.
5. The discussion section is too elaborate and needs more focus and direction
6. Overall, the study has good illustrations and proper methodology but poor English makes the manuscript hard to follow and understand. Addressing the above concerns will make manuscript suitable for Cancers
Author Response
Congratulations on the successful completion and presentation of results of the study titled 'Arytenoid Fixation in Laryngeal Cancer: Radiological Pictures and Clinical Correlations with Respect to Conservative Treatments'. The presentation of results in the manuscript is very good, but there are a few concerns to be addressed before the manuscript becomes suitable for publication in Cancers.
1. Introduction: The figure in the introduction section could have another image which can show the position of the crico-arytenoid unit in the human body. From which the CAU could be shown as a magnified version.
Author’s response: We are grateful to the Reviewer for this suggestion. A general view of the larynx in the sagittal plane was added as the left panel of figure 1
2. The grammar and the construction of some sentences needs to be corrected in the introduction section. For ex: "This retrospective mono-institutional study, conducted on a consecutive series of 29 patients affected by T3-T4a untreated LC, had as unique objective the attempt to identify some homogeneous subcategory in the large group of lesions determining arytenoid fixation, in order to understand which cases could be liable of conservative laryngeal treatment with sufficient security."
Author’s response: We are sorry. A complete check has been performed throughout the manuscript to improve grammar and syntax.
3. Figure 2,3,4 and 5: Labeling the figure with major parts will make the image more readable and easy to understand. without which the images are not identifiable.
Author’s response: We thank the Reviewer for this suggestion. We have modified the figure labelling: under each of them, and the caption is now more visible and recognizable.
4. The manuscript has various sections where English grammar has to be corrected to increase the quality of the content. For example: "About the infraglottic evolution, the chosen cut-off, meaning a subglottic extension < 10 mm from at the midline of the free edge of TVC ensures the fact that the tumor does not interested massively these anatomic structures yet". These are some examples of poor English in the manuscript.
Author’s response: Again, we apologize. As stated at point 2, a complete check has been performed throughout the manuscript to improve grammar and syntax.
5. The discussion section is too elaborate and needs more focus and direction
Author’s response: We thank the Reviewer for this advice. We have tried to focus more on some crucial parts of the discussion.
6. Overall, the study has good illustrations and proper methodology but poor English makes the manuscript hard to follow and understand. Addressing the above concerns will make manuscript suitable for Cancers
Author’s response: We hope the complete revision of syntax could adequately address this flaw.
Round 2
Reviewer 1 Report
The authors have addressed all my concerns.